# Norwegian moose CWD induces clinical disease and neuroinvasion in gene-targeted mice expressing cervid S138N prion protein

Maria Immaculata Arifin[1,☉], Samia Hannaoui[1,☉], Raychal Ashlyn Ng[1], Doris Zeng[1], Irina Zemlyankina[1], Hanaa Ahmed-Hassan[1,2], Hermann M. Schatzl[1,3,4], Lech Kaczmarczyk[5], Walker S. Jackson[5], Sylvie L. Benestad[6], Sabine Gilch[1,3,4]*

1 Faculty of Veterinary Medicine, University of Calgary, Calgary, Canada, 2 Zoonoses Department, Faculty of Veterinary Medicine, Cairo University, Giza, Egypt, 3 Hotchkiss Brain Institute, University of Calgary, Calgary, Canada, 4 Snyder Institute for Chronic Diseases, University of Calgary, Calgary, Canada, 5 Linköping University, Linköping, Sweden, 6 Norwegian Veterinary Institute, Ås, Norway

☉ These authors contributed equally to this work.
* sgilch@ucalgary.ca

**Data Availability Statement:** All data are available in the manuscript and supplementary information.

## Abstract

Chronic wasting disease (CWD) is a prion disease affecting deer, elk and moose in North America and reindeer, moose and red deer in Northern Europe. Pathogenesis is driven by the accumulation of PrP^Sc, a pathological form of the host's cellular prion protein (PrP^C), in the brain. CWD is contagious among North American cervids and Norwegian reindeer, with prions commonly found in lymphatic tissue. In Nordic moose and red deer CWD appears exclusively in older animals, and prions are confined to the CNS and undetectable in lymphatic tissues, indicating a sporadic origin.

We aimed to determine transmissibility, neuroinvasion and lymphotropism of Nordic CWD isolates using gene-targeted mice expressing either wild-type (138SS/226QQ) or S138N (138NN/226QQ) deer PrP. When challenged with North American CWD strains, mice expressing S138N PrP did not develop clinical disease but harbored prion seeding activity in brain and spleen. Here, we infected these models intracerebrally or intraperitoneally with Norwegian moose, red deer and reindeer CWD isolates. The moose isolate was the first CWD type to cause full-blown disease in the 138NN/226QQ model in the first passage, with 100% attack rate and shortened survival times upon second passage. Furthermore, we detected prion seeding activity or PrP^Sc in brains and spinal cords, but not spleens, of 138NN/226QQ mice inoculated intraperitoneally with the moose isolate, providing evidence of prion neuroinvasion. We also demonstrate, for the first time, that transmissibility of the red deer CWD isolate was restricted to transgenic mice overexpressing elk PrP^C (138SS/226EE), identical to the PrP primary structure of the inoculum.

Our findings highlight that susceptibility to clinical disease is determined by the conformational compatibility between prion inoculum and host PrP primary structure. Our study indicates that neuroinvasion of Norwegian moose prions can occur without, or only very limited, replication in the spleen, an unprecedented finding for CWD.

**Funding:** Alberta Prion Research Institute (https://albertainnovates.ca/alberta-prion-research-institute/; Grant #201800003 and #212200712 to SG, #201600023 to SG and HMS), Natural Sciences and Engineering Research Council Canada (https://www.nserc-crsng.gc.ca/index_eng.asp; Grant #RGPIN/05309 to SG), Margaret Gunn Endowment for Animal Research (https://research.ucalgary.ca/conduct-research/funding/apply-grants/internal-grants/margaret-gunn-endowment-animal-research-mgear) to SG, Parks Canada Agency (https://parks.canada.ca/agence-agency) to SG, and Genome Canada (https://genomecanada.ca/;Grant #10205 to SG). SG received salary support through the Canada Research Chairs program (https://www.chairs-chaires.gc.ca/home-accueil-eng.aspx). The funders had no role in study design, data collection and analysis, decision to publish, or preparation of the manuscript.

**Competing interests:** The authors have declared that no competing interests exist.

## Author summary

Chronic wasting disease (CWD) is a prion disease of cervids that is expanding its global footprint. The pathogenesis of prion disease is driven by the accumulation of PrP$^{Sc}$, a misfolded isoform of the cellular prion protein (PrP$^C$). CWD prion strains from North America are lymphotropic, while Norwegian moose and red deer prions are not, and therefore considered non-contagious, sporadic CWD forms.

We studied the propagation of Norwegian CWD prions in gene-targeted mice carrying cervid PrP$^C$ variants. We reveal that the Norwegian moose isolate induces clinical disease in mice expressing a PrP$^C$ variant previously shown to only display subclinical infection upon challenge with North American CWD. We report the first instance of red deer CWD transmission exclusively to mice overexpressing elk PrP$^C$.

Notably, our findings suggest a neuroinvasion route for Norwegian moose CWD prions that potentially bypasses spleen replication, underscoring the complexity of prion disease transmission, and the need for continued research into the behavior of prions across different species and protein variants.

## Introduction

Chronic wasting disease (CWD) is the prion disease of free-ranging and captive cervid populations in the United States, Canada, South Korea, Norway, Sweden, and Finland [1–3]. The disease is primarily caused by the aggregation and accumulation of PrP$^{Sc}$, a misfolded and infectious isoform of the host-expressed cellular prion protein (PrP$^C$). PrP$^{Sc}$ is thought to be the main, if not the only, component of prions [4,5]. CWD prions can replicate and accumulate in peripheral tissues of infected animals, particularly in the lymphatic system. Infected animals shed infectious prions into the environment through saliva, blood, urine, and feces [6–8], and environmental prion contamination can facilitate prolonged indirect transmission [9–11]. CWD in North America is highly contagious, and prevalence has reached up to ~19% within endemic regions and as high as ~87% in localized areas [12].

Several distinct CWD prion strains have been identified in North America [13–17]. They are all lymphotropic, contagious, and considered to be acquired by infection. By contrast, Norwegian CWD has distinct features; CWD in moose and red deer was identified in very old animals, with PrP$^{Sc}$ detectable in the brain but not in lymphatic tissues, while lymphotropism was only evident in cases of reindeer CWD [18–21]. This gave rise to the hypothesis that CWD in Norwegian moose and red deer represent sporadic, atypical forms of CWD, reminiscent of atypical/Nor98 scrapie and atypical H- and L-BSE strains [22–24]. On the host's side, previous studies have suggested that animals harboring PrP$^C$ allelic variants were less prone to contracting CWD compared to their wild-type counterparts [25–27]. However, certain prion strains have been shown to break these genotypic transmission barriers, e.g., the H95$^+$ and 116AG CWD strains from white-tailed deer were able to produce terminal prion disease in Tg60 mice expressing the 96S deer PrP$^C$, a mouse model resistant to transmission from other CWD isolates [14,16].

The objective of our study was to delve deeper into the characteristics of Norwegian CWD prion isolates, particularly their propagation dynamics in hosts that carry cervid PrP$^C$ variants known for their reduced susceptibility to CWD, e.g., S138N in reindeer/caribou. We were also interested in exploring their ability to infiltrate the brain following infection from peripheral sites. To this end, we characterized the propagation of three Norwegian CWD isolates, i.e., the M-NO3 moose isolate, R-NO16 reindeer isolate, and H-NO1 red deer isolate in gene-targeted

mice expressing either the wild-type deer representative for mule deer, white-tailed deer and reindeer/caribou (138SS/226QQ) or the 138NN (138NN/226QQ) PrP$^C$ variant found in caribou [28,29], or transgenic mice overexpressing elk (138SS/226EE) PrP$^C$ [30–32]. Our findings reveal that the M-NO3 moose isolate was able to induce clinical disease in gene-targeted mice expressing the 138NN-PrP$^C$ variant, an experimental host that previously only showed subclinical infection, i.e., prions were detected by *in vitro* amplification, but infected mice did not display clinical signs, upon infection with North American CWD isolates [29]. In addition, while M-NO3 did not elicit clinical disease in our mouse models following intraperitoneal inoculation, we detected prion seeding activity using RT-QuIC in several mouse brains and spinal cords, as well as proteinase K (PK)-resistant PrP$^{Sc}$ by western blot in the spinal cord of one *Prnp*.Cer.138NN mouse. Notably, there was no prion seeding activity detected in their spleens. This suggests a route of neuroinvasion for M-NO3 prions that may bypass or only transiently involve splenic prion replication. Furthermore, we report the novel finding that the H-NO1 red deer isolate can transmit disease exclusively to transgenic mice overexpressing elk PrP$^C$.

## Results

### Propagation of Norwegian CWD isolates in *Prnp*.Cer.Wt mice

Prior to inoculation, brain homogenates (BH) from R-NO16, M-NO3 [20], and H-NO1 [21] were tested for prion seeding activity in RT-QuIC using mouse recombinant PrP as a substrate and compared based on endpoint dilution and lag phase. Endpoint dilution is the highest sample dilution resulting in a positive reaction [33]. Ten percent BH from R-NO16 and M-NO3 were still positive for seeding activity when diluted up to $2 \times 10^{-4}$, while H-NO1 was positive only up to $2 \times 10^{-3}$ dilution, suggesting lower amounts of seeding-capable prions in the H-NO1 brain (**Fig 1**). Lag phase represents the time required to cross the threshold [33]. R-NO16 had the shortest lag phase with ~5 hours reaction time, followed by M-NO3 at ~10 hours, and H-NO1 at ~15 hours (**Fig 1**). This suggests that R-NO16 had the highest prion seeding activity, followed by M-NO3 and H-NO1.

Gene-targeted mice expressing wild-type deer PrP (*Prnp*.Cer.Wt), were inoculated intracerebrally (i.c.) or intraperitoneally (i.p.) with R-NO16, M-NO3 or H-NO1. Mice inoculated i.c. with R-NO16 and M-NO3 reached terminal prion disease with average survival times of 628.4 ± 25.4 days post-inoculation (dpi) and 700 ± 108.5 dpi, respectively (**Table 1**). Mice inoculated with R-NO16 displayed signs similar to those inoculated with a Canadian reindeer isolate [29], designated R-CA1 here onwards, the most apparent being ataxia, low-profile walking, and lethargy, while groups inoculated with M-NO3 mostly displayed signs of intense hyperactivity, loss of balance, and pronounced kyphosis. However, neither isolate resulted in a full attack rate, with R-NO16 giving a 71.4% and M-NO3 a 66.7% attack rate (**Table 1**). Mice inoculated i.c. with H-NO1 did not develop clinical disease up to 820 dpi (**Table 1**). Upon i.p. inoculation, only mice inoculated with R-NO16 reached terminal disease at 719.4 ± 30.7 dpi with 87.5% attack rate, while those inoculated i.p. with M-NO3 did not develop clinical signs of prion disease up to the experimental endpoint of 779 dpi (**Table 1**). Of note, experimental endpoints among gene-targeted mice were selected according to the natural lifespan of mouse models and to be comparable between groups of mice with different genotypes inoculated with the same CWD isolate, e.g., *Prnp*.Cer.Wt and *Prnp*.Cer.138NN mice inoculated i.p. with M-NO3 (**Table 1**).

Mice that exhibited clinical prion signs were euthanized upon terminal disease (*see materials and methods*) and were confirmed to harbor PK-resistant prions (PrP$^{res}$) in their brains by western blot analysis (**Fig 2A and 2B**). PrP$^{res}$ patterns, following first passage, were compared to original isolates from Norway used for inoculation, and to previously characterized CWD isolates from North America (**Fig 2D**). Norwegian isolates had low to moderate seeding

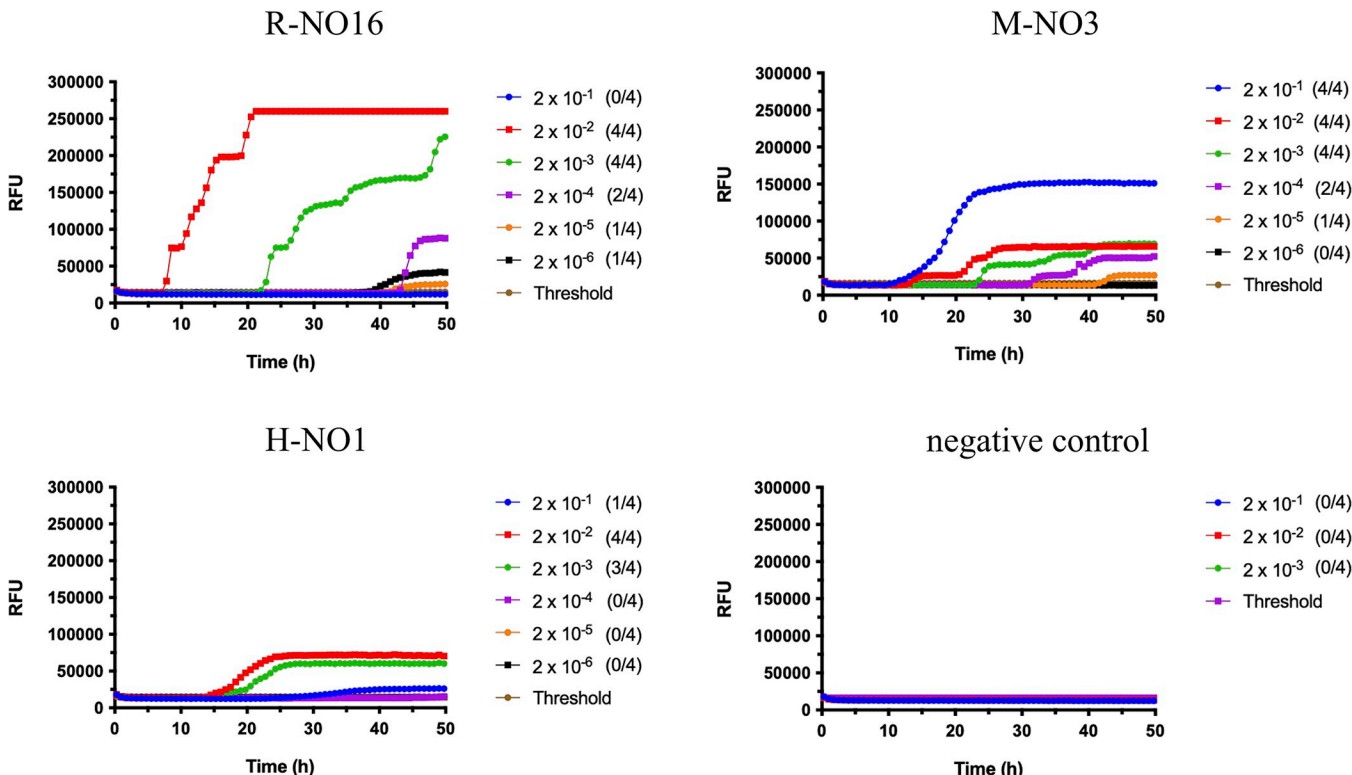

**Fig 1. Comparison of Norwegian isolates seeding activity in RT-QuIC.** Ten percent brain homogenates were subjected to serial dilutions from $2 \times 10^{-1}$ to $2 \times 10^{-6}$ in RT-QuIC seed dilution buffer. Samples were considered positive when a minimum of two out of four wells crossed the threshold relative fluorescence unit (RFU). Threshold is the average RFU of all negative control reactions plus five times their standard deviation. Negative control was a CWD-negative mule deer. R-NO16 and M-NO3 were positive for prion seeding activity up to $2 \times 10^{-4}$ dilution (upper panels), and H-NO1 up to $2 \times 10^{-3}$ dilution (bottom left panel). The y-axis represents the RFU, and the x-axis represents time in hours (h). Mouse recombinant PrP was used as substrate for the RT-QuIC reactions.

activity in RT-QuIC, and only M-NO3 harbored detectable amounts of PrP$^{res}$ by western blot. Despite similar seeding activity, R-NO16 did not show PrP$^{res}$, possibly due to a higher amount of PK-sensitive PrP$^{Sc}$ compared to M-NO3 (**Figs 2D and A in S1 Text**). Overall, brain homogenates from mice inoculated with M-NO3 harbored PrP$^{res}$ of lower molecular weight, with a pattern comparable to that of the original M-NO3 isolate. PrP$^{res}$ in brain homogenates of mice infected with R-NO16 appears similar to R-CA1, an experimental isolate from reindeer [34], and two isolates from white-tailed deer consisting of different strains (Wisc-1 and 116AG; 14,16). CWD from elk harboring strain CWD2 [17] exhibits PrP$^{res}$ with a molecular weight that is intermediate between that of M-NO3 and other North American CWD strains (WTD-Wisc-1, WTD-116AG and R-CA1). Additionally, the molecular weight of this PrP$^{res}$ differs from any PrP$^{res}$ detected in gene-targeted mice (**Figs 2 and A in S1 Text**) Mice euthanized due to reasons unrelated to prion disease or due to reaching our experimental endpoint (*see materials and methods*) did not contain levels of PrP$^{res}$ detectable by western blots in their brains (**Fig 2A–2C**). However, some of these PrP$^{res}$-negative mouse brains were positive for seeding activity in RT-QuIC (**Table 2** and **Fig B in S1 Text**), suggesting that they harbor either very low amounts of prions, or PK-sensitive prions. Spleens from these mice were analyzed by RT-QuIC and groups inoculated with R-NO16 were positive, independent of the inoculation route, and, to a lesser extent, those inoculated with H-NO1 (**Table 2**). Interestingly, *Prnp*.Cer. Wt mice inoculated i.p. with M-NO3 were the only ones that did not exhibit any seeding activity in either their brains or spleens (**Table 2**). Additionally, brains and spleens of mice

**Table 1. Survival times of *Prnp*.Cer.Wt, *Prnp*.Cer.138NN and TgElk mice following intracerebral (i.c.) and intraperitoneal (i.p.) inoculation with Norwegian CWD isolates.**

| | | | | Average survival times ± *SD* in days ($N_1/N_{total}$) | | |
|---|---|---|---|---|---|---|
| Inoculum | Tissue | passage | roi | *Prnp*.Cer.Wt | *Prnp*.Cer.138NN | TgElk |
| R-NO16 | Brain | 1 | i.c. | 628.4 ± 25.4 (5/7) | 734 (5/5) [b] | na |
| Reindeer | | | | 637 (1/7) [a], 733 (1/7) [b] | | |
| | | 1 | i.p. | 719.4 ± 30.7 (7/8) | 734 (4/4) [b] | na |
| | | | | 703 (1/8) [a] | | |
| M-NO3 | Brain | 1 | i.c. | 700 ± 108.5 (4/6) | 565, 656 (2/5) | 258 (4/4) [b] |
| Moose | | | | 623, 635 (2/6) [a] | 720 (1/5) [a], 756, 832 (2/5) [b] | |
| | | 1 | i.p. | 297, 653 (2/5) [a], 779 (3/5) [b] | 600, 651, 720 (3/5) [a], 774, 782 (2/5) [b] | na |
| | | 2 | i.c. | na | 360.6 ± 34.7 (9/9) | na |
| H-NO1 | Brain | 1 | i.c. | 572, 656, 720 (3/5) [a], 818, 820 (2/5) [b] | 714 (1/4) [a], 784, 798, 832 (3/4) [b] | 166 (1/4) |
| Red deer | | | | | | 257 (3/4) [b] |
| | | 1 | i.p. | na | 629, 629, 705 (3/3) [a] | na |
| | | 2 | i.c. | na | na | 149.3 ± 44.5 (4/5) |
| | | | | | | 315 (1/5) [b] |

[a] Humane endpoint euthanasia with no clinical signs or confirmed prion disease, defined in methods.

[b] Experimental endpoint euthanasia with no clinical disease, defined as $\geq 700$ dpi for mice inoculated with R-NO16, $\geq 750$ dpi for mice inoculated with M-NO3 and H-NO1, and $\geq 250$ dpi for the TgElk mouse line independent of inoculum (see methods). $N_1/N_{total}$ = number of mice euthanized / number of mice per group; *SD* = standard deviation; roi = route of inoculation; i.c. = intracerebral; i.p. = intraperitoneal; na = not inoculated.

inoculated i.p. were examined for abnormal PrP deposits by immunohistochemistry and only those inoculated with R-NO16 were found positive (**Fig C in S1 Text**). In contrast to the same mouse model inoculated with R-CA1 [29], mice inoculated with R-NO16 did not harbor PrP deposits in their hippocampus (**Fig C in S1 Text**).

The absence of clinical disease, PrP$^{res}$, abnormal PrP deposits, and prion seeding activity in the brains of *Prnp*.Cer.Wt mice inoculated i.p. with M-NO3, as well as the absence of seeding activity in the spleens of both i.c. and i.p. inoculated groups, provide evidence that M-NO3 prions are not able to propagate in lymphatic tissue, regardless of inoculation route, in animals expressing wild-type deer PrP$^C$ (138SS/226QQ).

## Norwegian moose CWD isolate causes clinical disease in *Prnp*.Cer.138NN mice

A gene-targeted mouse line harboring caribou-*Prnp* homozygous for asparagine at codon 138 (*Prnp*.Cer.138NN), was inoculated i.c. and i.p. with the R-NO16, M-NO3 and H-NO1 CWD isolates from Norway. The *Prnp*.Cer.138NN mouse line was resistant to clinical prion disease upon inoculation with various CWD isolates and strains from North America, including reindeer and white-tailed deer prions [29].

Here, the M-NO3 moose isolate was able to break this transmission barrier and produced clinical disease in two out of the five *Prnp*.Cer.138NN mice inoculated i.c. (40% attack rate; **Table 1**). PrP$^{res}$ with a low molecular weight similar to that of the original M-NO3 isolate, but slightly different between the two mice, was detectable in their brains upon PK digestion and western blotting (**Figs 2B, 2D and A in S1 Text**). One other sample had a signal at around 25 kDa, which was confirmed to be unspecific/non-PrP related after deglycosylation. No clinical signs of CWD infection or PrP$^{res}$ were detected in the brains of *Prnp*.Cer.138NN mice inoculated i.p. with M-NO3 or inoculated by either route with R-NO16 or H-NO1 (**Table 1 and Fig 2A–2C**).

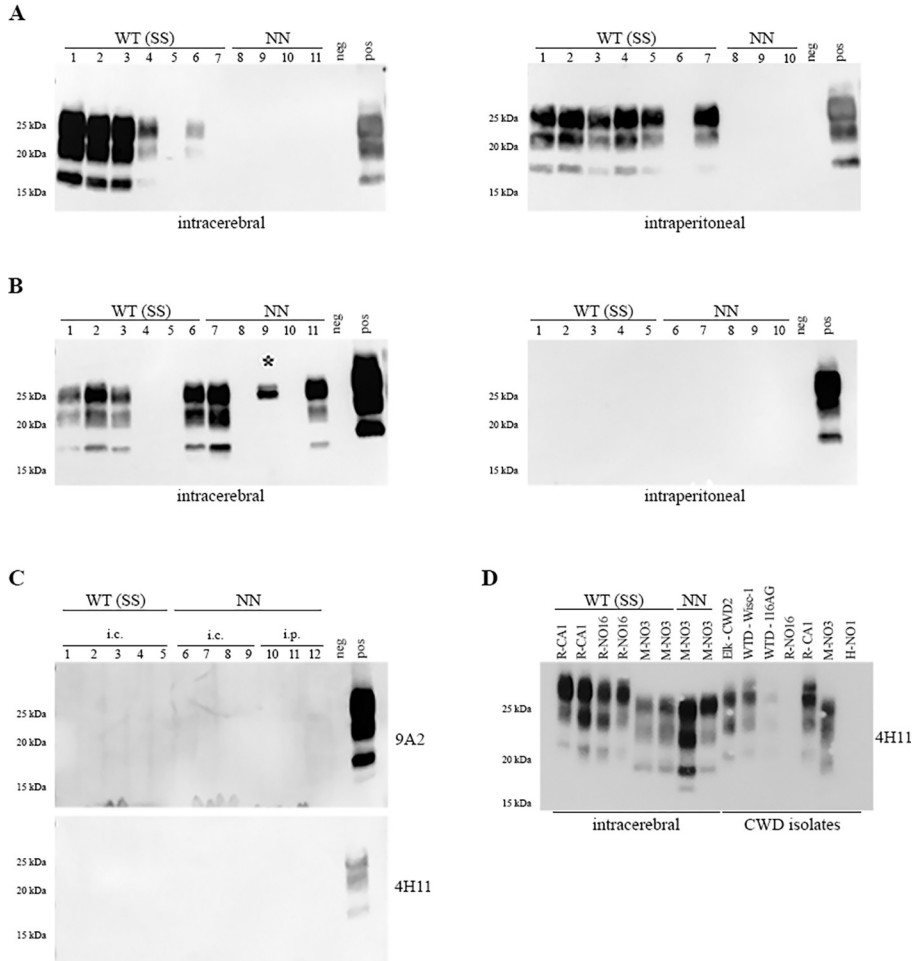

**Fig 2. PrP<sup>res</sup> detection in *Prnp*.Cer.Wt and *Prnp*.Cer.138NN brain homogenates of mice inoculated i.c. and i.p. with R-NO16 (A), M-NO3 (B), and H-NO1 (C) isolates.** Ten percent brain homogenates were digested with 50 μg/mL PK for one hour at 37°C. For (**A**) and (**B**), the anti-PrP antibody used was 4H11 (1:500), and for (**C**) it was 9A2 (1:1000, upper panel) and 4H11 (1:500, lower panel). (**D**) PrP<sup>res</sup>-positive brain homogenates of gene-targeted mice as indicated inoculated i.c. with R-CA1, R-NO16, and M-NO3 were analyzed by western blot along with CWD isolates from North America (Elk-CWD2, WTD-Wisc-1, WTD-116AG, R-CA1) and Norway (R-NO16, M-NO3, H-NO1) digested with 50 μg/ml of PK using anti-PrP antibody 4H11 (1:500). R-NO16 and H-NO1 did not harbor detectable amounts of PrP<sup>res</sup>. Samples from Asterisk (*) in Fig 2B indicates non-specific signal. WT (SS) = *Prnp*.Cer.Wt; NN = *Prnp*.Cer.138NN; neg = non-inoculated *Prnp*.Cer.138NN mouse brain; pos = PrP<sup>res</sup>-positive *Prnp*.Cer.Wt control brain; kDa = kilodalton; i.c. = intracerebral; i.p. = intraperitoneal.

When the PrP<sup>res</sup>-negative brains of *Prnp*.Cer.138NN mice and their spleens were tested for prion seeding activity in RT-QuIC, between 50 and 75% of the brains from i.c.-inoculated animals were positive, regardless of inoculum, whereas all spleens were negative (**Table 2**). This is different from North American CWD prions, which had detectable seeding activity in the spleen upon i.c. inoculation in this mouse model [29].

## Divergent PrP<sup>res</sup> patterns in brains of *Prnp*.Cer.138NN mice inoculated i.c. with M-NO3

PrP<sup>res</sup> profiles of R-CA1, R-NO16, and M-NO3 in *Prnp*.Cer.Wt mice, and M-NO3 in *Prnp*.Cer.138NN mice were further analyzed by western blot and compared using various monoclonal anti-PrP antibodies (**Fig 3**). Unglycosylated PrP<sup>res</sup> in *Prnp*.Cer.Wt mouse brains inoculated

**Table 2. Number of mouse brains and spleens positive for PrPres on WB and/or prion seeding activity in RT-QuIC per total number of samples tested within each respective group.**

| | | | Number of positive samples / number of samples tested | | | | | | | | | | | | | |
| --- | --- | --- | --- | --- | --- | --- | --- | --- | --- | --- | --- | --- | --- | --- | --- | --- |
| | | | *Prnp*.Cer.Wt | | | | | | | *Prnp*.Cer.138NN | | | | | | |
| | | | Brain | | | | Spleen | | | Brain | | | | Spleen | | |
| Inoculum | Tissue | roi | WB | RT-QuIC | | | RT-QuIC | | | WB | RT-QuIC | | | RT-QuIC | | |
| | | | PrPres | -1 | -2 | -3 | -1 | -2 | -3 | PrPres | -1 | -2 | -3 | -1 | -2 | -3 |
| R-NO16 | Brain | i.c. | 5/7 | – | – | – | 2/7 | 3/7 | 3/7 | 0/4 | 0/4 | 2/4 | 2/4 | 0/4 | 0/4 | 0/4 |
| Reindeer | | | | | | | | | | | | | | | | |
| | | i.p. | 6/7 | – | – | – | 2/6 | 4/6 | 4/6 | 0/3 | 1/3 | 1/3 | 0/3 | 2/3 | 2/3 | 2/3 |
| M-NO3 | Brain | i.c. | 4/6 | – | – | – | 0/5 | 0/5 | 0/5 | 2/5 | 2/5 | 3/5 | 1/5 | 0/5 | 0/5 | 0/5 |
| Moose | | | | | | | | | | | | | | | | |
| | | i.p. | 0/5 | 0/5 | 0/5 | 0/5 | 0/4 | 0/4 | 0/4 | 0/5 | 2/5 | 4/5 | 1/5 | 0/4 | 0/4 | 0/4 |
| H-NO1 | Brain | i.c. | 0/5 | 0/5 | 5/5 | 0/5 | 0/3 | 0/3 | 1/3 | 0/4 | 0/4 | 3/4 | 2/4 | 0/4 | 0/4 | 0/4 |
| Red deer | | | | | | | | | | | | | | | | |
| | | i.p. | na | na | na | na | na | na | na | 0/3 | 0/3 | 0/3 | 0/3 | 0/3 | 0/3 | 0/3 |

The numbers -1, -2, and -3 refer to 2 x $10^{-1}$, 2 x $10^{-2}$, and 2 x $10^{-3}$ RT-QuIC seed dilutions, respectively. Samples were considered positive in RT-QuIC when a minimum of two out of four replicates crossed the threshold relative fluorescence unit (RFU, see methods). Brain homogenates positive for PrPres on WB (Fig 2) were not tested in RT-QuIC, except for *Prnp*.Cer.138NN mouse brains inoculated i.c. with M-NO3. Brain homogenates from *Prnp*.Cer.Wt mice inoculated i.c. and i.p. with R-NO16 and i.c. with M-NO3 that were negative for PrPres on WB (Fig 2) were tested in RT-QuIC but not included in this table (data shown in Fig B in S1 Text).
roi = route of inoculation; WB = western blot; i.c. = intracerebral; i.p. = intraperitoneal;– = not tested; na = not inoculated.

with R-CA1 and R-NO16 had similar molecular weights at ~19 kDa, while PrPres from mice inoculated with M-NO3 migrated slightly faster at ~18 kDa (Fig 3). Interestingly, of the two PrPres-positive *Prnp*.Cer.138NN mice inoculated i.c. with M-NO3 prions, one mouse had an unglycosylated PrPres band with a similar size as its *Prnp*.Cer.Wt counterparts, while the other showed a slightly faster migrating band at ~17 kDa (Fig 3). Upon probing with monoclonal antibodies raised against the N-terminus of PrP, 9A2 and 12B2, none of the PrP glycoforms in the sample with the ~17 kDa PrPres (white arrowheads in Fig 3A) seem to be detected, and only faint signals were detectable in the three samples harboring the ~18 kDa PrPres band (Fig 3A). After deglycosylation, PrPres from all four mice inoculated with M-NO3 were still detected with 4H11, but not with 12B2 (Fig 3B). Overexposure of the 12B2 blot revealed faint PrPres bands for the ~18 kDa samples, but none for the ~17 kDa sample (Fig 3B). This indicates that the N-terminal cleavage site of PrPres is different in these two mouse brains. Interestingly, in three out of the four *Prnp*.Cer.Wt mice, an additional faint PrPres band at approximately ~16 kDa was detectable with 8H4, which was absent in the samples from *Prnp*.Cer.138NN mice (Fig 3C). These findings confirm previously described strain differences between R-NO16 and M-NO3 [18,35], and indicate the divergence of M-NO3 into PrPres signatures with ~18 or ~17 kDa unglycosylated fragments in *Prnp*.Cer.138NN mice.

Next, we performed a second passage of M-NO3 in *Prnp*.Cer.138NN mice. We used a pool of brain homogenates from the two PrPres-positive *Prnp*.Cer.138NN mice inoculated with M-NO3 for i.c. passage (n = 9). All mice in this group developed terminal, clinical prion disease, with a significantly shortened survival time of 360.6 ± 34.7 dpi compared to the first passage with 565 and 656 dpi, respectively (Fig 4A and Table 2). Brain homogenates of all mice were positive for PrPres in western blot (Figs 4B and D in S1 Text). We compared the PrPres pattern and reactivity with different anti-PrP antibodies in brain homogenates of mice upon 2nd passage with those observed in the 1st passage. Interestingly, all brain homogenates from

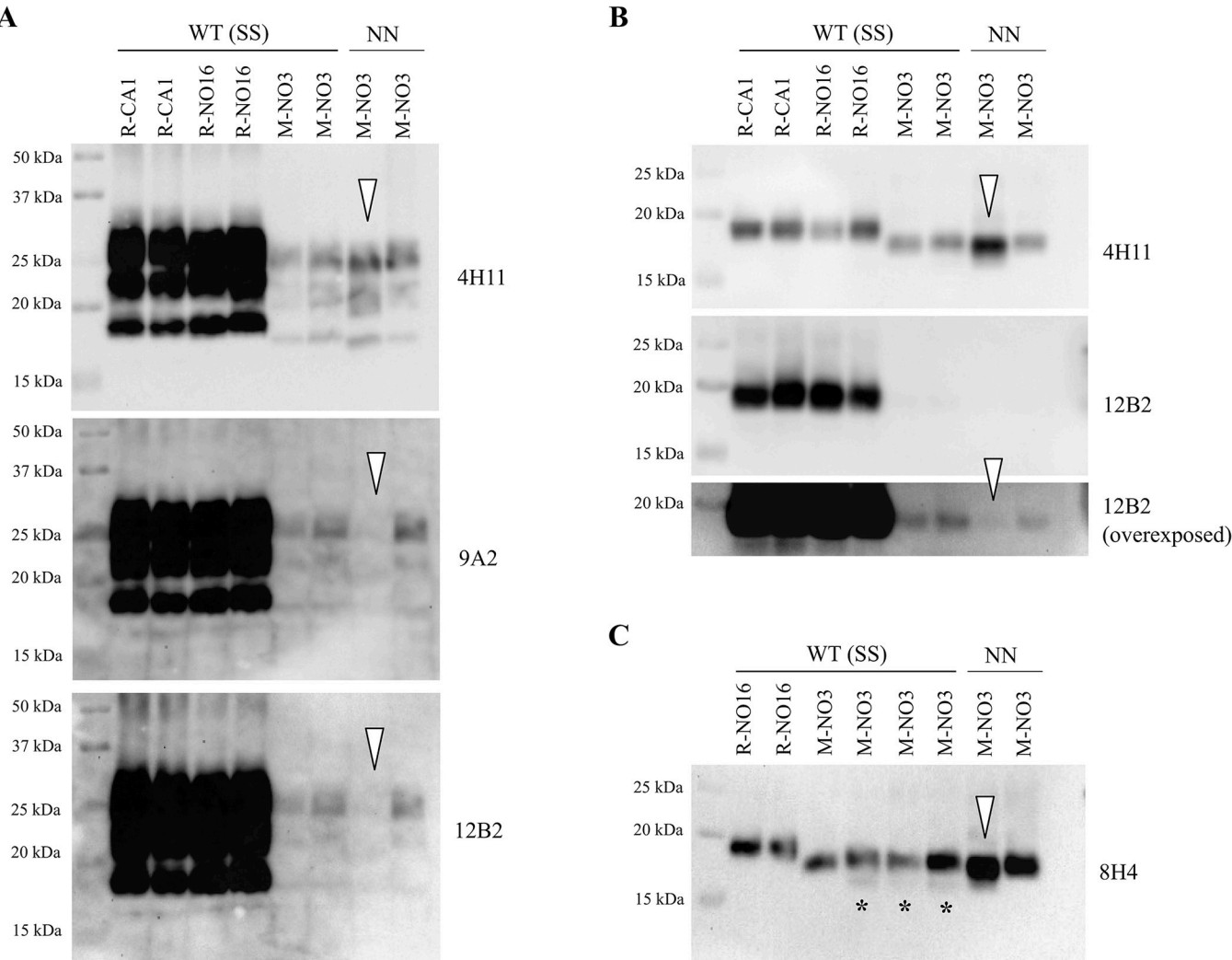

**Fig 3. Comparison of PrP^res in brain homogenates of *Prnp*.Cer.Wt and *Prnp*.Cer.138NN mice inoculated i.c. with R-CA1, R-NO16, or M-NO3.** (**A**) Ten percent brain homogenates were digested with 50 μg/mL PK for one hour at 37°C. Anti-PrP antibodies used were 4H11 (1:500), 9A2 (1:1000) and 12B2 (1:1000). (**B**) PK-digested brain homogenates were then subjected to deglycosylation using the PNGase-F enzyme and probed with 4H11 and 12B2 and (**C**) with 8H4 (1:2000). Open arrowheads indicate the *Prnp*.Cer.138NN sample with a ~17 kDa PrP^res band that is not detectable by 12B2. Asterisks (*) indicate the ~16 kDa PrP^res band detected with 8H4 in *Prnp*.Cer.Wt samples. WT (SS) = *Prnp*.Cer.Wt; NN = *Prnp*.Cer.138NN; kDa = kilodalton.

2^nd passage had PrP^res similar to the higher molecular weight pattern in 1^st passage, which was detectable with monoclonal antibody 12B2 (**Figs 4B and D in S1 Text**).

## Prion transport to the CNS in *Prnp*.Cer.138NN mice inoculated i.p. with M-NO3

Upon i.p. inoculation, brains and spleens of H-NO1-inoculated *Prnp*.Cer.138NN mice did not harbor detectable prion seeding activity. As expected, seeding activity was detectable in both brains and spleens of mice infected i.p. with R-NO16 (**Table 2**).

Notably, *Prnp*.Cer.138NN mice inoculated i.p. with M-NO3 showed prion seeding activity using RT-QuIC, in their brains and spinal cords, but not their spleens (**Figs 5, E in S1 Text and Table 2**). Serial protein misfolding cyclic amplification (sPMCA) up to 5 rounds was consistently negative when brain or spleen homogenates were analyzed using brain homogenates of either *Prnp*.Cer.Wt or *Prnp*.Cer.138NN as a substrate (representative result shown in **Fig F**

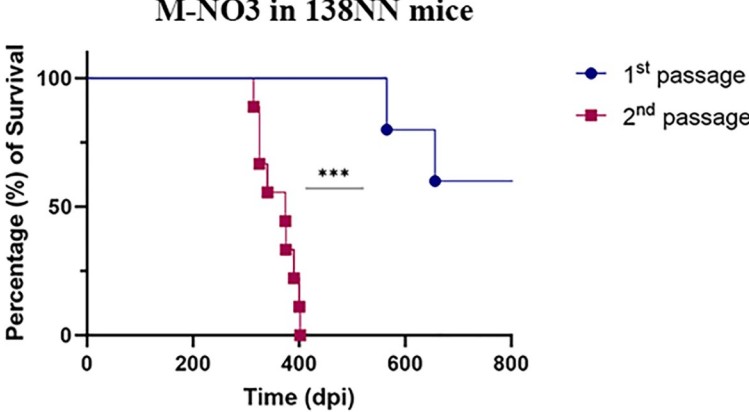

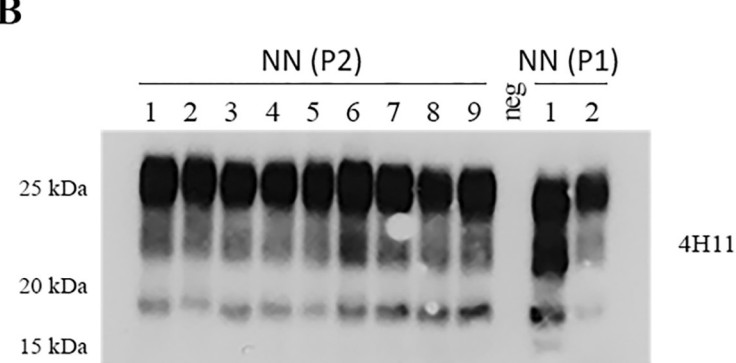

**Fig 4. Second passage of M-NO3 in *Prnp*.Cer.138NN mouse model. (A)** Comparison of survival times of *Prnp*.Cer.138NN inoculated with M-NO3 (1st passage; n = 5; attack rate = 40%; see **Table 1**). Equal volumes of 10% brain homogenates of two mice with terminal prion disease in the 1st passage were pooled and used for the 2nd passage using a group of n = 9 *Prnp*.Cer.138NN mice for i.c. inoculation. Survival times are depicted as Kaplan-Meyer curve, and the log-rank test was used for statistical analysis. *** *p*-value = 0.0006. **(B)** Brain homogenates of 1st and 2nd passage were digested with 50 μg/ml of PK and subjected to western blot using anti-PrP antibody 4H11 (1:500), and 12B2 (**Fig D in S1 Text**).

in **S1 Text**), indicating a lower sensitivity of detection by serial PMCA compared to RT-QuIC under the experimental conditions used here. Notably, PrP^res was detectable by western blot in the spinal cord sample of the i.p. inoculated *Prnp*.Cer.138NN mouse displaying the highest seeding activity. This finding supports the RT-QuIC results (**Figs 5 and G in S1 Text**) and demonstrates the transport of M-NO3 prions to the CNS. These findings suggest that M-NO3 prions are capable of neuroinvasion upon peripheral infection in *Prnp*.Cer.138NN, but not *Prnp*.Cer.Wt mice. This was specific to M-NO3 prions, since *Prnp*.Cer.138NN mice inoculated i.p. with H-NO1 prions did not produce the same outcome (**Table 2**).

## Norwegian red deer isolate is transmissible to transgenic mice expressing elk PrP^C

The Norwegian red deer isolate H-NO1 was inoculated i.c. into transgenic mice expressing elk PrP^C [30,31], on the basis that this red deer was homozygous for glutamic acid (E) at *Prnp* codon

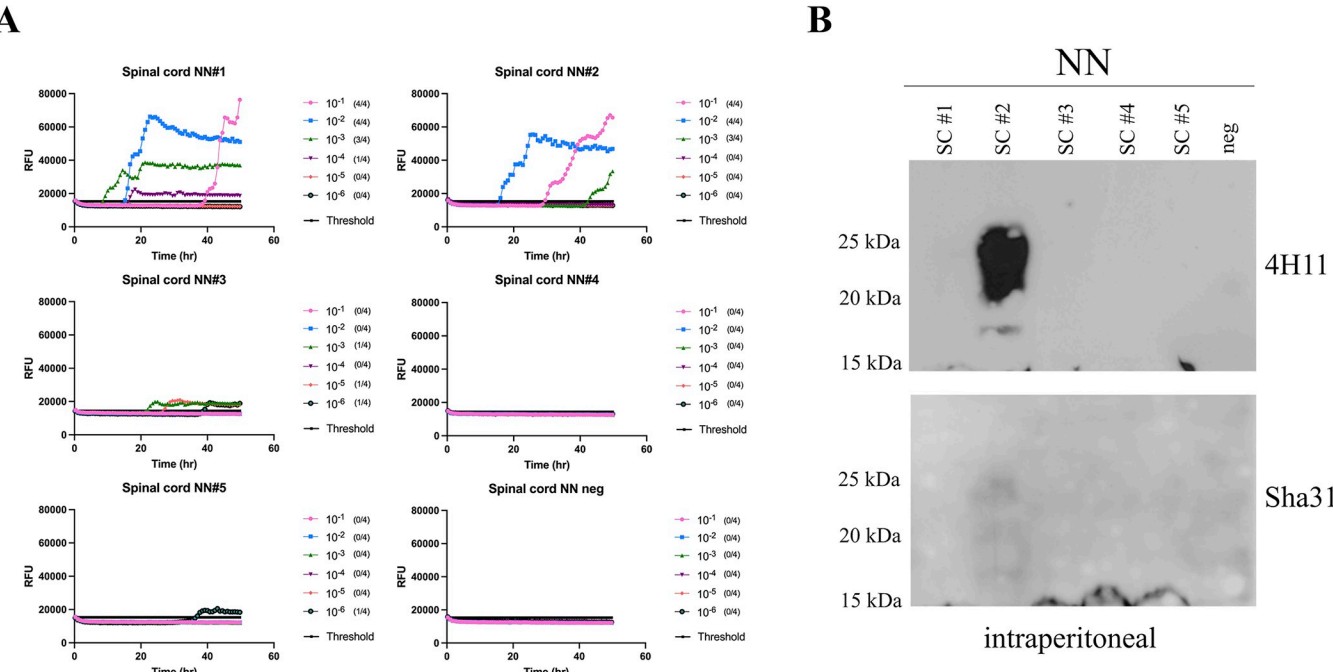

**Fig 5. RT-QuIC and western blot analysis of spinal cord homogenates of *Prnp*.Cer.138NN mice inoculated i.p. with M-NO3. (A)** Ten percent spinal cord homogenates were subjected to serial dilutions from $2 \times 10^{-1}$ to $2 \times 10^{-6}$ in RT-QuIC seed dilution buffer. Samples were considered positive when a minimum of two out of four wells crossed the threshold relative fluorescence unit (RFU). Threshold is the average RFU of all negative control reactions plus five times their standard deviation. Negative control was a naïve spinal cord homogenate of a *Prnp*.Cer.138NN mouse. The y-axis represents the RFU, and the x-axis represents time in hours (h). Mouse recombinant PrP was used as substrate for the RT-QuIC reactions. **(B)** Spinal cord homogenates were digested with 50 μg/ml of PK and subjected to western blot analysis using anti-PrP antibodies 4H11 (1:500) and Sha31 (1:10,000); see also overexposed western blot in **Fig G in S1 Text**).

226 [21], which is identical to elk *Prnp* (226EE). The H-NO1 red deer isolate was able to transmit disease in TgElk mice, albeit at a low attack rate of 25%, where only one out of the four mice inoculated developed disease at 166 dpi (**Table 1**). PrP^res was present in the brain of this singular mouse (**Fig 6**). Upon second passage, the attack rate increased to 80% (four out of five mice) while the average survival time remained comparable at 149.3 ± 44.5 dpi (**Table 2**). The four terminally ill TgElk mice from the second passage accumulated PrP^res in their brains (**Fig 6**). Of note, M-NO3 was also inoculated into TgElk for comparison, and none of the mice developed disease up to 258 dpi (**Table 1**). The effective propagation of H-NO1 prions in TgElk mice, along with the subclinical presence of H-NO1 prions with seeding activity in the brains and spleens of *Prnp*.Cer.Wt, and in the brains of *Prnp*.Cer.138NN mice, suggests that codon 226 may influence the transmission of Norwegian red deer prions. Specifically, homozygosity for glutamic acid at this codon (226EE) appears to facilitate the development of full-fledged CWD, whereas the presence of glutamine (226QQ) may only allow for subclinical prion propagation upon first passage.

## Discussion

We have previously shown that gene-targeted mice expressing cervid PrP^C homozygous for asparagine at codon 138 (*Prnp*.Cer.138NN) and inoculated with North American CWD isolates do not develop clinical CWD, but harbor seeding-capable prions, mostly in their spleens [29]. Here we report that the Norwegian moose isolate M-NO3 was able to break the genotypic transmission barrier into *Prnp*.Cer.138NN mice upon i.c. inoculation, resulting in clinical CWD and accumulation of PrP^res in their brains.

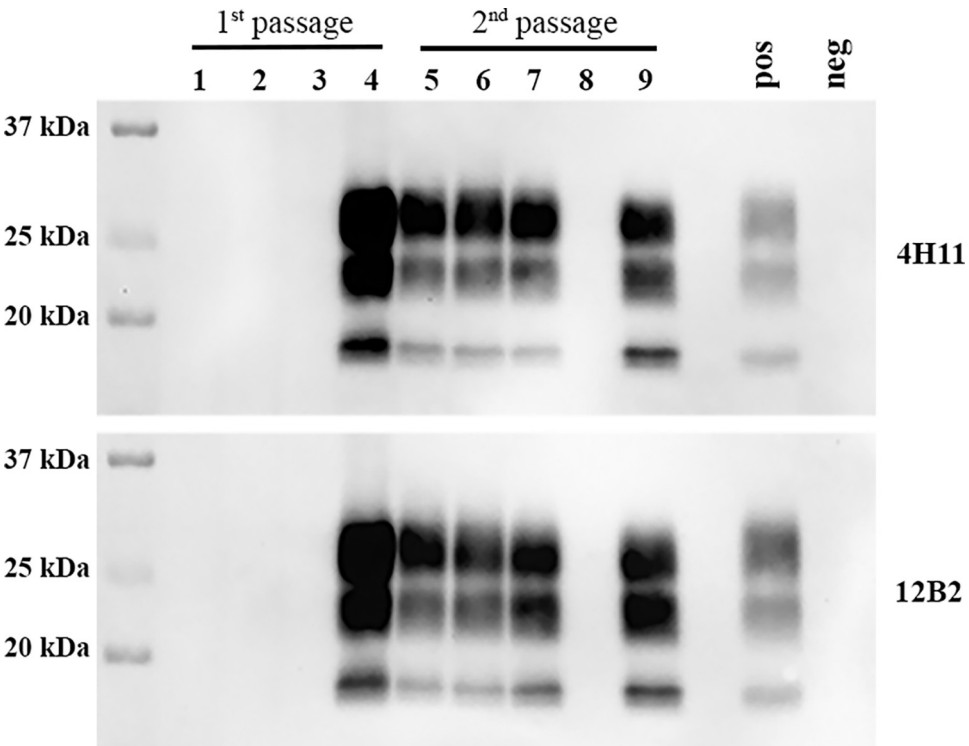

**Fig 6. PrP^res detection in brain homogenates of TgElk mice inoculated with H-NO1.** Ten percent brain homogenates were digested with 50 μg/mL PK for one hour at 37°C. Anti-PrP antibodies used were 4H11 (1:500, upper panel) and 12B2 (1:1000, lower panel). Lanes 1–4 corresponds to TgElk mice inoculated i.c. with the H-NO1 brain homogenate. Lanes 5–9 correspond to TgElk mice inoculated with brain homogenate from the TgElk in lane 4. pos = PrP^res-positive TgElk control; neg = non-inoculated TgElk; kDa = kilodalton.

M-NO3 was shown to propagate in bank voles and several transgenic mouse models, and to possess different properties compared to North American and other Norwegian CWD isolates [18,35]. Here, in all *Prnp*.Cer.Wt and one of the clinical *Prnp*.Cer.138NN mice, M-NO3 retained an ~18 kDa unglycosylated PrP^res fragment, detectable by the N-terminal 12B2 antibody [20,35]. Interestingly, although we did not consistently observe other unglycosylated PrP^res fragments reported previously [18,20], we mainly detected a ~17 kDa fragment in the second PrP^res-positive *Prnp*.Cer.138NN mouse, not recognized by 12B2 antibody. Similar to previous reports [14,36], this implies the existence of a minor sub-strain within M-NO3, which selectively propagated in one *Prnp*.Cer.138NN mouse, while *Prnp*.Cer.Wt mice and the other clinically affected *Prnp*.Cer.138NN mouse only propagated the major strain. Alternatively, the variant with the lower molecular weight might have evolved upon passage in the *Prnp*.Cer.138NN mice. However, in 2^nd passage, the majority of PrP^res detected in the mouse brains appeared comparable to the ~18 kDa variant detectable by 12B2, indicating that the ~17 kDa variant might be less stable upon passage, or replicating less efficiently, considering that the group of mice was inoculated with a pool of the two PrP^res-positive brain homogenates from the 1^st passage.

We previously demonstrated that North American CWD strains propagated in spleens of both *Prnp*.Cer.Wt and *Prnp*.Cer.138NN mice inoculated i.c. and i.p. [29]. Propagation of the Norwegian moose isolate M-NO3 was confined to the brain in both mouse lines, with no seeding activity detectable in the spleen by RT-QuIC, even upon i.p. inoculation, at the endpoints of our experiments. This corroborates previous reports that M-NO3 is not lymphotropic

[18,20] and extends these findings for the first time to peripheral infection. Intraperitoneal inoculation of both *Prnp*.Cer.Wt and *Prnp*.Cer.138NN mice with M-NO3 did not result in clinical disease up to ~750 dpi, but *Prnp*.Cer.138NN mice had seeding activity in their brains and spinal cords, as well as detectable PrP$^{res}$ in the spinal cord of one mouse. Prion seeding activity was not evident in *Prnp*.Cer.Wt mice infected i.p. with M-NO3, and we argue that 138N-PrP$^C$ might be more efficiently converted by M-NO3 prions than 138S-PrP$^C$. This is supported by the shorter survival times of i.c.-inoculated *Prnp*.Cer.138NN mice, compared to their wild-type (138SS) counterparts.

Our results suggest that M-NO3 prions are capable of neuroinvasion from peripheral sites of infection. The lack of detectable prion seeding activity in the spleens of these mice suggests that replication in the spleen either does not occur during pathogenesis and therefore, does not play a role in neuroinvasion, or that replication in the spleen is transient and may not necessarily contribute to neuroinvasion. The absence of prion seeding activity in spleen contrasts with findings in the majority of CWD cases and our gene-targeted mouse models. In these instances, prion seeding activity and/or PrP$^{res}$ are consistently detected in lymphatic tissue, e.g., tonsil, lymph node, spleen, at the terminal stage of the disease [29,37,38]. This discrepancy suggests variations in prion/strain behavior and disease progression across different models. Furthermore, prions are frequently detected in the lymphatic tissues of infected cervids but not their brain, and not vice versa [37,39]. Overall, these reports suggest that in CWD, lymphatic tissue replication precedes neuroinvasion and that detectable levels of prions persist in lymphatic tissue until terminal disease. Additional experiments to study kinetics of prion replication and transport in different tissues throughout the incubation period in *Prnp*.Cer.138NN mice are needed to rule out the possibility of early or rapid prion clearance in the spleen following neuroinvasion [40–42]. It is important to note though that a minor pathway of neuroinvasion independent of the spleen or the immune system with direct spread from the peritoneum via visceral nerve fibers has been described [43,44], consistent with our detection of prion seeding activity and PrP$^{res}$ in spinal cords. These data do not contradict the proposition that CWD in Norwegian moose started spontaneously at an advanced age but may suggest that infected animals propagate prions at peripheral sites other than lymphatic tissues [41]. It also indicates that environmental contamination from deceased infected moose might result in transmission to other cervids, even though in an inefficient manner and restricted to certain PrP$^C$ genotypes [9,10,45–47]. The *Prnp* of European fallow deer (*Dama dama*) is not polymorphic at codon 138 and encodes asparagine (138NN), in contrast to serine in the wild-type PrP of the majority of cervid species, and harbors glutamate at position 226 (226EE; 27,48). Fallow deer were resistant to mule deer CWD upon natural exposure, but not to intracerebral inoculation [47–49]. It remains to be determined whether fallow deer are more susceptible to peripheral infection with Norwegian moose CWD, or whether they would be protected by the glutamate at residue 226 (226EE), as shown in gene-targeted mice expressing cervid 138SS/226EE (elk) PrP$^C$ [18]. Furthermore, the presence of prions within uncharacterized peripheral reservoirs in the Norwegian moose might provide a source of prion exposure if consumed [50–52].

For R-NO16, the PrP$^{res}$ profile was identical to R-CA1 upon passaging into *Prnp*.Cer.Wt mice. However, R-NO16 resulted in prolonged incubation periods, lower attack rates, as well as the absence of hippocampal abnormal PrP deposition, confirming previous findings that R-NO16 and R-CA1 are different CWD strains [18,35]. Of note, a higher attack rate in animals inoculated i.p. was observed. Although this could be due to enhanced peripheral replication resulting in more 'efficient' prion neuroinvasion or selection of more 'replicable' prions [53,54], it could also simply be due to individual mouse differences, given the small difference in the number of animals testing positive, i.e., 7 out of 8 in i.p.- vs 5 out of 7 in i.c-inoculated groups. Upon i.c. and i.p. inoculation with R-NO16, most *Prnp*.Cer.Wt mice showed positive

seeding activity in their spleens. However, this was not the case for *Prnp*.Cer.138NN mice inoculated i.c., indicating a possible restriction in prion spread within these animals when inoculated with R-NO16. This might be due to inefficient anterograde transport, very low prion replication in the brain, or prion degradation [40–42]. It is crucial to note that this observation is unique to the Norwegian CWD isolates examined in this study, as the same mouse line readily replicated North American CWD prions in their spleens following i.c. inoculation [29]. The majority of the *Prnp*.Cer.138NN mice inoculated i.p. with R-NO16 harbored seeding activity in their spleens and brains, demonstrating that R-NO16 can replicate within the spleens of this mouse line, and is transported efficiently to the brain.

We also report for the first time that the Norwegian red deer isolate H-NO1 is infectious and transmissible. Identical PrP$^{Sc}$ and PrP$^C$ genotypes were required for H-NO1 to induce clinical disease. The differential transmissibility of M-NO3 and H-NO1 to TgElk indicates that these two isolates harbor different CWD strains; however, further characterization is needed for confirmation. A genotype match between the host PrP$^C$ and incoming prions often results in the most efficient prion disease transmission [55–58]. While H-NO1 (226EE) successfully transmitted full-fledged CWD to TgElk mice (226EE), albeit with incomplete attack rates, it failed to cause clinical prion disease in both *Prnp*.Cer.Wt (226QQ) and *Prnp*.Cer.138NN (226QQ) mice. It is important to note that 226QQ hosts tend to generate fewer stable strains than those expressing 226EE [17], suggesting more stable H-NO1 prion propagation in TgElk mice, which was confirmed upon second passage. Nevertheless, we detected prion seeding activity in the brains of both mouse lines inoculated i.c., and in the spleen of one *Prnp*.Cer.Wt mouse. Peripheral infection of *Prnp*.Cer.138NN mice with H-NO1 was even less effective, with no detectable prions in their brains and spleens. Unfortunately, we did not have the *Prnp*.Cer. Wt group inoculated i.p. to directly compare, due to limited resources. Nevertheless, these findings still support previous reports that H-NO1 is poorly, if at all, lymphotropic, indicated by the absence of abnormal PrP deposits in this red deer's lymph nodes and tonsils [21].

In summary, we confirmed that the transmission of CWD prions is not merely a matter of host genotype. Rather, it is the unique traits of each prion strain guiding the course of transmission. One of our most intriguing findings is that neuroinvasion of Norwegian moose CWD prions can occur possibly without, or only with minor/transient, involvement of prion replication in the spleen, which has not been reported for CWD prions [37–39,59,60]. Even though it is, arguably, possible that spleen prion clearance occurred early on, the absence of detection upon terminal disease is still an uncommon trait of CWD prions [37,38], which also suggests the absence of prion dissemination from the CNS to the periphery and subsequent shedding. These observations broaden the spectrum of atypical features of Norwegian moose CWD prions. They underscore the intricate interplay between prion strains and host PrP genotypes that define transmission barriers, thus reminding us of the diversity and complexity inherent in the world of prions.

## Materials and methods

### Ethics statement

This study was performed under animal care protocols AC19-0047 and AC22-0015 approved by the University of Calgary Health Sciences Animal Care Committee and in compliance with the guidelines of the Canadian Council for Animal Care.

### CWD isolates

Reindeer isolate R-NO16 (ID #18-CD2788) was from a free-ranging reindeer in the Nordfjella region and was not characterized previously. R-NO16 had the A/C PrP genotype, i.e., alleles A (Ser225)/C (deletion) and 109KK [61]. Moose isolate M-NO3 (ID #17-CD11399) was

described previously [18,20,35]. Both the reindeer and moose isolates were homozygous for serine and glutamine at codons 138 (138SS) and 226 (226QQ), respectively [18,20,35,61]. Red deer isolate H-NO1 (ID #17-CD14501) was from a free-ranging animal from the Gjemnes region, Norway, and was homozygous for serine and glutamic acid at codons 138 (138SS) and 226 (226EE), respectively [21]. WTD-Wisc-1, WTD-116AG, and R-CA1 isolates were described previously [14,16,34]. Brains were homogenized as 10% (w/v) in sterile 1X PBS using the gentleMACS Dissociator (Miltenyi Biotec) and stored at -80˚C.

## Bioassays

Gene-targeted mice expressing deer PrP$^C$ with either serine (*Prnp*.Cer.Wt; 138SS/226QQ) or asparagine (*Prnp*.Cer.138NN; 138NN/226QQ) at residue 138 and transgenic TgElk mice expressing 2.5-fold elk PrP$^C$ (138SS/226EE) have been described [29–32].

Eight female mice per group were i.c. inoculated between six to eight weeks of age with 20 μl of 1% BH or i.p. inoculated with 100 μl of 5% BH, as previously described [14]. For 2nd passage of M-NO3 in *Prnp*.Cer.138NN mice, equal volumes of the two PrP$^{res}$ positive brain homogenates were pooled for i.c. inoculation. Mice were monitored weekly for signs of prion disease and then daily at the onset of at least one confirmatory sign, including rigid tail, hind limb clasping, front and/or hind limb ataxia, loss of righting reflex, circling, occasional tremors, and weight loss. At that point, mice were euthanized by isoflurane overdose followed by cardiac perfusion [29]. Survival time or dpi was defined as the number of days from the date of inoculation to the date of euthanasia. Humane endpoint euthanasia was performed on sick or injured mice showing no clinical signs of prion disease. Animal experiments were terminated at ~700 to 750 dpi for gene-targeted mice, and ~250 dpi for TgElk mice to prevent non-prion related health issues with this mouse line. Mice euthanized prior to 100 dpi due to humane reasons or found dead were excluded from the analyses.

## Mouse tissue collection and homogenization

Brains and spleens were harvested, immediately frozen on dry ice, and stored at -80˚C, or fixed in formalin. Frozen brains and spleens were homogenized as 20% and 10% (w/v) homogenates, respectively, using the FastPrep-24 Tissue Homogenizer system (MP Biomedicals, Canada).

## RT-QuIC

For RT-QuIC, we tested 10% (w/v) brain, spinal cord, or spleen homogenates diluted in a buffer containing 0.1% SDS and 1X N2 media supplement, in serial dilutions from $2 \times 10^{-1}$ up to $10^{-6}$, with four replicates each. Recombinant mouse PrP (aa 23–231) preparation, RT-QuIC reactions, and analyses were performed as previously described [33].

## Serial Protein Misfolding Cyclic Amplification (sPMCA)

sPMCA was performed as described previously [29]. Briefly, brains from non-inoculated *Prnp*.Cer.Wt, and *Prnp*.Cer.138NN were prepared as 10% (w/v) BH in cold PMCA conversion buffer (4 mM EDTA, 1% Triton X-100 and 1 tablet cOmplete Protease Inhibitor Mini [Roche] in 1X PBS, pH of 7.4). Samples were tested at a $2 \times 10^{-2}$ dilution, with R-CA1-infected *Prnp*.Cer.Wt mouse brain homogenate serving as a positive control. Sonication was set for 30 s at 375–395 W followed by 29.5 min rest and run for 24 h for a total of 48 sonication-rest cycles, corresponding to one round of PMCA. For sPMCA, 10 μl of sample was transferred into 90 μl of fresh PMCA substrate. A total of five rounds were conducted for each substrate.

## SDS-PAGE and western blotting

Mouse BH (20% w/v) was diluted at a 1:1 ratio in 2X lysis buffer (100 mM of Tris-HCl pH 7.5, 100 mM of NaCl, 10 mM of EDTA, 1% Triton-X, 1% sodium deoxycholate, $H_2O$). Fifty μg/ml PK was added into the mix, incubated at 37˚C for one hour, followed by boiling for 10 min at 95˚C in SDS sample buffer. To remove N-linked glycans, PK-digested samples were incubated with PNGase-F enzyme for 24 h at 37˚C. Samples were subjected to SDS-PAGE and immunoblotting using anti-PrP antibodies 4H11, 9A2 (aa 102–104, sheep PrP numbering), 12B2 (aa 93–97, sheep PrP numbering), 8H4 (aa 145–180, human PrP numbering) and Sha31 (aa 145–152, human PrP numbering) as previously described [29].

## Immunohistochemistry

Immunohistochemistry was performed as previously described [16,29]. Briefly, formalin-fixed brains and spleens embedded in paraffin were mounted on positively-charged slides, deparaffinized, autoclaved in antigen retrieval buffer, incubated in 98% formic acid, and treated with 4 M of guanidine thiocyanate. Abnormal PrP deposits were detected using the BAR224 anti-PrP antibody (aa 141–151, human PrP numbering).

## Supporting information

**S1 Data. Raw data file.**
(XLSX)

**S1 Text. Supplementary Information. Fig A. PrP<sup>res</sup> in brain homogenates of gene-targeted mice and various CWD isolates.** PrP^res^-positive brain homogenates of gene-targeted mice as indicated inoculated i.c. with R-CA1, R-NO16, and M-NO3 were analyzed by western blot along with CWD isolates from North America (Elk-CWD2, WTD-Wisc-1, WTD-116AG, and R-CA1) and Norway (R-NO16, M-NO3, and H-NO1) digested with 50 μg/ml of PK using anti-PrP antibodies 12B2. R-NO16 and H-NO1 did not harbor detectable amounts of PrP^res^.
**Fig B. Seeding activity in PrP<sup>res</sup>-negative brains of *Prnp*.Cer.Wt mice inoculated with R-NO16 and M-NO3.** Left column shows RT-QuIC results from individual *Prnp*.Cer.Wt mice inoculated i.c. (n = 2) and i.p. (n = 1) with R-NO16 that were negative for PrP^res^ on western blotting (**Fig 2**). Right column shows RT-QuIC results from individual *Prnp*.Cer.Wt mice inoculated i.p. with M-NO3 (n = 2) that were negative for PrP^res^ on western blotting (**Fig 2**). Samples were considered positive when a minimum of two out of four reactions crossed the threshold relative fluorescence unit (RFU), indicated by the violet line. Positive dilutions are highlighted in black rectangles. The threshold is the average RFU of all negative control reactions plus five times their standard deviation. Negative control was a non-inoculated *Prnp*.Cer.Wt mouse brain. The y-axis represents the RFU, and the x-axis represents time in hours (h).
**Fig C. Representative immunohistochemistry of brain and spleen sections from gene-targeted mice inoculated i.p. with the R-NO16 and M-NO3 isolates.** Abnormal PrP deposits were detected in the frontal cortex, cerebellum, and spleen of the *Prnp*.Cer.Wt mouse inoculated with R-NO16 (row 3, highlighted with the black rectangle), but not M-NO3 (row 5). No abnormal PrP deposits were detected in *Prnp*.Cer.138NN mice tissues inoculated with both isolates (rows 2 and 4). No abnormal PrP deposits were detected in the hippocampus of all mice tested. Detection of abnormal PrP deposits was performed using the anti-PrP antibody BAR224 (1:2000) with hematoxylin counterstain. **Fig D. Western blot analysis of *Prnp*.Cer.138NN brain homogenates inoculated with M-NO3.** Brain homogenates of M-NO3 1<sup>st</sup> and 2<sup>nd</sup> passage were digested with 50 μg/ml of PK and subjected to western blot analysis using anti-PrP antibody 12B2. Non-infected *Prnp*.Cer.138NN brain homogenate served as a

negative control. **Fig E. RT-QuIC analysis of PrP<sup>res</sup>-negative brains and spleens of *Prnp*. Cer.138NN mice inoculated i.p. with M-NO3. (A)** Brain (n = 5) and (B) spleen (n = 4) homogenates of *Prnp*.Cer.138NN mice inoculated i.p. with M-NO3 that were negative for PrP<sup>res</sup> in western blot (**Fig 2**) were serially diluted and analyzed by RT-QuIC using mouse recombinant PrP as a substrate. Samples were considered positive when a minimum of two out of four reactions crossed the threshold relative fluorescence unit (RFU), indicated by the black line. The threshold is the average RFU of all negative control reactions plus five times their standard deviation. Negative control was non-inoculated *Prnp*.Cer.138NN mouse brain and spleen, respectively. The y-axis represents the RFU, and the x-axis represents time in hours (h). **Fig F. Serial PMCA of brain and spleen homogenates of *Prnp*.Cer.138NN mice inoculated i.p. with M-NO3.** 2 x 10–2 dilutions of brain (n = 5) and spleen (n = 3) homogenates of *Prnp*.Cer.138NN mice inoculated with M-NO3 were subjected to 5 rounds of PMCA using naïve brain homogenates of either *Prnp*.Cer.Wt (**A**) or *Prnp*.Cer.138NN (**B**) mice as a substrate. sPMCA products of rounds 4 and 5 were digested with PK (50 μg/ml) for 1 hour and analyzed by western blot using anti-PrP mAb 4H11 (1:500). Naïve brain homogenate was used as a negative control, brain homogenate of *Prnp*.Cer.Wt mice inoculated with R-CA1 served as a positive control. **Fig G. Western blot analysis of spinal cord homogenates.** Spinal cord homogenates of *Prnp*.Cer.138NN mice inoculated i.p. with M-NO3 were digested with 50 μg/ml of PK and analyzed by western blot using anti-PrP antibody Sha31 (1:10,000). This is an overexposed version of the western blot shown in **Fig 4B** (**lower panel**). (DOCX)

## Acknowledgments

Portions of the paper were developed from the thesis of M.I.A. We would like to thank the staff and veterinarians at the Prion-Virology Animal Facility and Clara Christie Centre for Mouse Genomics for excellent animal care. We would also like to acknowledge Keegan McDonald for helping with tissue homogenization, Yo-Ching Cheng for producing the recombinant PrP used in our RT-QuIC runs, and Dr. Gordon Mitchell (Canadian Food Inspection Agency Ottawa, Canada) for providing us with the North American reindeer isolate. We are grateful to Dr. Debbie McKenzie (University of Alberta, Canada) for sharing the TgElk mouse model and the WTD-Wisc-1 isolate, to Dr. Trent Bollinger (Canadian Wildlife Health Cooperative, Canada) for sharing the WTD-116AG isolate, and to Dr. Stefanie Czub (Canadian Food Inspection Agency, Lethbridge, Canada) for providing the Elk-CWD2 isolate.

## Author Contributions

**Conceptualization:** Maria Immaculata Arifin, Samia Hannaoui, Sylvie L. Benestad, Sabine Gilch.

**Data curation:** Maria Immaculata Arifin, Samia Hannaoui, Raychal Ashlyn Ng, Doris Zeng, Irina Zemlyankina, Hanaa Ahmed-Hassan.

**Formal analysis:** Maria Immaculata Arifin, Samia Hannaoui.

**Funding acquisition:** Hermann M. Schatzl, Sabine Gilch.

**Investigation:** Maria Immaculata Arifin, Samia Hannaoui, Raychal Ashlyn Ng, Doris Zeng, Irina Zemlyankina, Hanaa Ahmed-Hassan.

**Methodology:** Maria Immaculata Arifin, Samia Hannaoui, Raychal Ashlyn Ng, Doris Zeng, Irina Zemlyankina, Hanaa Ahmed-Hassan, Lech Kaczmarczyk, Walker S. Jackson.

**Project administration:** Sabine Gilch.

**Resources:** Lech Kaczmarczyk, Walker S. Jackson, Sylvie L. Benestad, Sabine Gilch.

**Supervision:** Maria Immaculata Arifin, Samia Hannaoui, Sabine Gilch.

**Validation:** Maria Immaculata Arifin, Samia Hannaoui, Sabine Gilch.

**Visualization:** Maria Immaculata Arifin, Samia Hannaoui.

**Writing – original draft:** Maria Immaculata Arifin, Sabine Gilch.

**Writing – review & editing:** Maria Immaculata Arifin, Samia Hannaoui, Raychal Ashlyn Ng, Doris Zeng, Irina Zemlyankina, Hanaa Ahmed-Hassan, Hermann M. Schatzl, Lech Kaczmarczyk, Walker S. Jackson, Sylvie L. Benestad, Sabine Gilch.

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
