## [Decision Letter · Decision Letter 0]

22 Dec 2023

Dear Dr. Gilch,

Thank you very much for submitting your manuscript "Norwegian moose CWD induces clinical disease and spleen-independent neuroinvasion in gene-targeted mice expressing cervid S138N prion protein" for consideration at PLOS Pathogens. As with all papers reviewed by the journal, your manuscript was reviewed by members of the editorial board and by several independent reviewers. In light of the reviews (below this email), we would like to invite the resubmission of a significantly-revised version that takes into account the reviewers' comments.

While Reviewers 1 and 3 were largely positive about the manuscript, Reviewer 2 had significant reservations about the data and asked for additional experiments. In your response, please be sure to address the comments of Reviewer 1, who was concerned that the conclusion that Norwegian moose CWD can be neuroinvasive in the absence of replication in the spleen was a bit strong in the absence of any timepoint analysis or longer incubation times. Please also be sure to address the concerns of Reviewer 2 with regard to further support for the RT-QuIC data in non-clinical brain and spleen and the addition of western blot data comparing PrPSc in the original inocula. Reviewer 2 also asked for a second passage in transgenic mice to further characterize and confirm the isolation of different strains. However, if these data are not readily available, this issue might be satisfactorily addressed by careful discussion of data already in the paper.

We cannot make any decision about publication until we have seen the revised manuscript and your response to the reviewers' comments. Your revised manuscript is also likely to be sent to reviewers for further evaluation.

Sincerely,

Suzette A. Priola

Guest Editor

PLOS Pathogens

Surachai Supattapone

Section Editor

PLOS Pathogens

Kasturi Haldar

Editor-in-Chief

PLOS Pathogens

orcid.org/0000-0001-5065-158X

Michael Malim

Editor-in-Chief

PLOS Pathogens

orcid.org/0000-0002-7699-2064

While Reviewers 1 and 3 were largely positive about the manuscript, Reviewer 2 had significant reservations about the data and asked for additional experiments. In your response, please be sure to address the comments of Reviewer 1, who was concerned that the conclusion that Norwegian moose CWD can be neuroinvasive in the absence of replication in the spleen was a bit strong in the absence of any timepoint analysis or longer incubation times. Please also be sure to address the concerns of Reviewer 2 with regard to further support for the RT-QuIC data in non-clinical brain and spleen and the addition of western blot data comparing PrPSc in the original inocula. Reviewer 2 also asked for a second passage in transgenic mice to further characterize and confirm the isolation of different strains. However, if these data are not readily available, this issue might be satisfactorily addressed by careful discussion of data already in the paper.

Reviewer's Responses to Questions

**Part I - Summary**

Reviewer #1: These study compares a North American strain of CWD to mouse, reindeer, and red deer isolates of CWD from Norway in gene targeted mice with various PRNPs. This work is interesting and important because it adds to the evidence that there are multiple CWD strains, including one that may have some overall features similar to Nor-98 scrapie or atypical BSE.

The authors compare interesting isolates of CWD in appropriate mouse models. There are a few weaknesses (detailed in the sections below) that would make it a better paper. In some instances, the claims made seem overstated for the work that was done, for example because the length of time that mice were allowed to incubate was short or because timepoint samples were not examined.

Reviewer #2: The main purpose of this study is to evaluate the transmission properties of three European CWD isolates. This was tested in three different animal models expressing deer, elk and caribou PrP, and two different routes of transmission (intra-cerebral and intra-peritoneal). The results show novel properties for these isolates, including host range, tissue tropism, and biochemical properties of the agent. In general, the study is informative by providing information of the host range of these isolates, and demonstrating the potential susceptibility of caribou to European CWD. By comparing with previous studies, they confirm that these European strains are different compared with their North American counterparts. As mentioned, the experiment is informative. However, the experiments are incomplete and no solid conclusions can be drawn with the current data. In addition, the experiment is incremental, building on previous characterizations already made for these isolates. These, and other weaknesses identified in this manuscript, are listed below.

Reviewer #3: This is a very detailed [a[er and should be published. I have no corrections

**Part II – Major Issues: Key Experiments Required for Acceptance**

Reviewer #1: Title and elsewhere: A claim is made that Norwegian moose CWD gets to the brain independent of the spleen, but no timepoint samples are analyzed and the potential for some degree of PrP Sc degradation is not addressed. This other work should be done or these claims should be softened and these other caveats should be explained somewhere in the manuscript.

Reviewer #2: 1. The current study is incremental, as it provides additional information on isolates that have been described in other publications.

2. Detection of prions in brain and spleen of non-clinical animals is provided by RT-QuIC. To provide more convincing evidence of subclinical infection, these samples should also be tested in a secondary method (e.g., PMCA, cell assays, bioassay).

3. Western blot of the original inocula used for this study should be included. The PrPSc generated in mice upon inoculation of the European CWD strains should be compared, side by side, with this inocula and with other CWD strains already characterized.

4. The potential isolation of different strains is a relevant result. As such, this needs to be further characterized, including second passages in transgenic mice. Although it is understandable that these experiments take long time, they are needed to draw solid conclusions (as presented and discussed by the authors).

Reviewer #3: I have no major issues.

**Part III – Minor Issues: Editorial and Data Presentation Modifications**

Reviewer #1: Abstract- a couple of suggestions that may be difficult depending on word limits-

Line 32-33- What do you mean by subclinical infection? Evidence of PrPSc, but not clinical signs at time of euthanasia?

Line 35: can you address the potential titer issue here? RT-QuIC suggests maybe we could have expected reindeer to incubate faster if amount only.

Line 52: again, what is subclinical infection to you? This can be misleading if mice were euthanized prior to the time that would be required to develop clinical signs.

Introduction- associated with paragraph startint at line 81-

Could you introduce what you are referring to for the various genotypes (wt, deer, caribou, elk) how they are the same/different and put the terminology here that will be used throughout because this isn't actually clearly explained until the methods, which occurs at the end.

Line 86: which deer?

Line 94: one potential place to address difference between spleen independent vs transient splenic replication.

Line 100: please add the substrate used here. It is not in the M&M and only appears in the figure legend.

Line 106-107: Is it an expectation that all of these will replicate equally in the substrate and the differences are due to amount of PrPSc present?

Line 118: Thoughts on the higher attack rate after IP inoculation (maybe a comment in the discussion)?

Line 120: Please comment on the incubation period allowed for the wt mice with the M-NO3 inoculum. This does not seem long enough ensure that all of these were negative since the IC incubation period was 700 +/- 108 days.

Line 129: (doesn't require a response) Western blot may not be sensitive to identify cases that would be detectable by other methods (ELISA, for example). It takes a relatively high OD/strong positive by ELISA before WBs are positive without some enrichment method. It would be better to call these non-detect rather than negative.

Line 137: ... activity in either their brains or spleens

Line 151: suggest omitting strong

Line 160: omit was able to break this transmission barrier and

Line 179: please add something to the title of this section, so it is clear that you're talking about WB's

Line 209: Is 258 days enough to make a claim? This isolate took 700 days in the other mouse strain.

Line 212-213: Worth looking at Angers, Science, 2010 and considering a reference as this codon has been identified with strain-associated replication differences.

Line starting 216: is there a way to add something about the inoculum (species of origin) from the previous studies, so the reader knows how to categorize the information?

Line 237: no clinical disease at what DPI?

line 239: again there is the potential that at some point clean up outpaces accumulation

Line 245: What about hematogenous spread?

Line 271: please clarify this sentence- you can transmit CWD prions (but not clinical disease)

Line 286: as above, is without involvement of the spleen is not the same as not being able to detect PrPSc at the endpoint.

Line 288: strains and genotypes that define. . .

Line 299: please describe this genotype in the same way as for the mouse strains. This is hard to follow.

Figure 4: how would these migrations compare to the reindeer and moose isolates in figure 3?

Reviewer #2: 1. M-NO3 is mentioned in the abstract. Specific information should be avoided in this section, as it is confusing at this point. A more general message, and no specific data (on a specific isolate) should be mentioned in this section.

2. In Table 1, Elk should be listed in the last column to agree with the flow of the manuscript.

3. Table 1. It is not clear why the animals noted with "b" are included in the table.

Reviewer #3: I have no minor issues.

PLOS authors have the option to publish the peer review history of their article (what does this mean?). If published, this will include your full peer review and any attached files.

Reviewer #1: No

Reviewer #2: No

Reviewer #3: No
---

## [Decision Letter · Decision Letter 1]

18 Jun 2024

Dear Dr. Gilch,

We are pleased to inform you that your manuscript 'Norwegian moose CWD induces clinical disease and neuroinvasion in gene-targeted mice expressing cervid S138N prion protein' has been provisionally accepted for publication in PLOS Pathogens.

Best regards,

Suzette A. Priola

Guest Editor

PLOS Pathogens

Neil Mabbott

Section Editor

PLOS Pathogens

Michael Malim

Editor-in-Chief

PLOS Pathogens

orcid.org/0000-0002-7699-2064

Reviewer Comments (if any, and for reference):

Reviewer's Responses to Questions

**Part I - Summary**

Reviewer #1: (No Response)

Reviewer #2: (No Response)

Reviewer #3: I have reviewed this article before and have no changes. It is a well written paper and should be published.

**Part II – Major Issues: Key Experiments Required for Acceptance**

Reviewer #1: (No Response)

Reviewer #2: (No Response)

Reviewer #3: (No Response)

**Part III – Minor Issues: Editorial and Data Presentation Modifications**

Reviewer #1: (No Response)

Reviewer #2: (No Response)

Reviewer #3: (No Response)

PLOS authors have the option to publish the peer review history of their article (what does this mean?). If published, this will include your full peer review and any attached files.

Reviewer #1: No

Reviewer #2: No

Reviewer #3: **Yes: **Terry Spraker

---

## [Editor Report · Acceptance letter]

22 Jun 2024

Dear Dr. Gilch,

We are delighted to inform you that your manuscript, "Norwegian moose CWD induces clinical disease and neuroinvasion in gene-targeted mice expressing cervid S138N prion protein," has been formally accepted for publication in PLOS Pathogens.

Best regards,

Michael Malim

Editor-in-Chief

PLOS Pathogens

orcid.org/0000-0002-7699-2064